# Repeated Drop-Weight Impact Testing of Fibrous Concrete: State-Of-The-Art Literature Review, Analysis of Results Variation and Test Improvement Suggestions

**DOI:** 10.3390/ma15113948

**Published:** 2022-06-01

**Authors:** Sallal R. Abid, Gunasekaran Murali, Jawad Ahmad, Thaar S. Al-Ghasham, Nikolai Ivanovich Vatin

**Affiliations:** 1Department of Civil Engineering, Wasit University, Kut 52003, Iraq; thaar@uowasit.edu.iq; 2Peter the Great St. Petersburg Polytechnic University, 195251 Saint Petersburg, Russia; murali_22984@yahoo.com (G.M.); vatin@mail.ru (N.I.V.); 3Department of Civil Engineering, Swedish College of Engineering and Technology Wah Cantt, Taxila 47040, Pakistan; jawadcivil13@scetwah.edu.pk

**Keywords:** repeated impact test, test improvement, normal distribution, Weibull distribution, fibrous concrete, results variation

## Abstract

The ACI 544-2R introduced a qualitative test to compare the impact resistance of fibrous concretes under repeated falling-mass impact loads, which is considered to be a low-cost, quick solution for material-scale impact tests owing to the simplified apparatus, test setup and procedure, where none of the usual sophisticated sensors and data acquisition systems are required. However, previous studies showed that the test results are highly scattered with noticeably unacceptable variations, which encouraged researchers to try to use statistical tools to analyze the scattering of results and suggest modifications to reduce this unfavorable disadvantage. The current article introduces a state-of-the-art literature review on the previous and recent research on repeated impact testing of different types of fibrous concrete using the ACI 544-2R test, while focusing on the scattering of results and highlighting the adopted statistical distributions to analyze this scattering. The influence of different mixture parameters on the variation of the cracking and failure impact results is also investigated based on data from the literature. Finally, the article highlights and discusses the literature suggestions to modify the test specimen, apparatus and procedure to reduce the scattering of results in the ACI 544-2R repeated impact test. The conducted analyses showed that material parameters such as binder, aggregate and water contents in addition to the maximum size of aggregate have no effect on the variation of test results, while increasing the fiber content was found to have some positive influence on decreasing this variation. The survey conducted in this study also showed that the test can be modified to lower the unfavorable variations of impact and failure results.

## 1. Introduction

Impacts from dropping objects or the collision of moving vehicles are types of accidental impact loads on horizontal and vertical structural members that are expected along their life spans, as heavy or light building units or objects can fall from higher altitudes during the construction period, while columns and walls of parking garages are expected to be repeatedly rammed by vehicles [1,2,3]. On the other hand, runways of airports are exposed to daily repeated impacts from the tires of landing plans, while offshore piers in harbor wharfs are also subjected to possible repeated impacts from hosted ships and sea waves [4,5]. Other possible sources of impact forces can be projectiles and fragmented parts from explosions in conflict areas or due to terrorist attacks [6,7]. The latter can be assessed using special high-velocity impact tests that simulate the effect of projectile hits, among which are the blast test [8,9,10,11,12] and projectile impact test [13,14,15,16,17,18]. On the other hand, for low-velocity impact tests, the ACI 544-2R [19] defined three major test procedures. The first is the instrumented falling impact test, while the second is the Charpy pendulum impact test [20,21,22,23]. The instrumented impact test provides reliable experimental explanations about the structural performance of structural members such as slabs and beams under drop-weight impact loads [24,25,26,27]. However, such a type of test requires large-scale testing machines and significantly costly specimens and instrumentations. On the other hand, a simple alternative test, which is the third impact test introduced by the ACI 544-2R [19], is the repeated drop-weight impact test. This test is very simple to perform and requires a cheap testing apparatus and no sophisticated instrumentations, which can be used as a low-cost, quick alternative to the drop-weight impact test to qualitatively compare the impact performance of different concrete mixtures.

The ACI 544-2R repeated impact test apparatus is simply composed of a 4.54 kg free dropping steel mass that is dropped on an approximately 150 mm diameter and 63.5 mm depth concrete disk specimen from a height of 457 mm. The impact mass is lifted manually by hand and left to be freely dropped by gravity from the specified height on a 63.5 mm diameter steel ball. The steel ball is hung on the center of the top surface of the specimen by a steel ring which is held by special steel framing plates to a stiff steel baseplate. The impact load from the falling mass is transferred to the central point of the specimen’s top surface through the steel ball. The baseplate is provided by special steel lugs that leave a 5 mm gap from the outer perimeter of the test specimen. The manual impacts are repeated on the specimen until a surface crack becomes visible. At this stage, the number of impact blows is recorded as the cracking impact number. The test is then continued until the crack is widened or the specimen is fractured so that it touches at least three of the four perimeter steel lugs. At this stage, the test is discontinued, and the number of impact blows is recorded as the failure impact number.

The ACI 544-2R introduced this procedure to evaluate the impact resistance of fibrous concrete. However, it is reported that the test is not intended to provide a quantitative evaluation of the impact strength or performance, where it can only be used as a qualitative comparison tool between different fibrous mixtures or with plain reference concrete. The test apparatus and procedure were first introduced by Schrader [28], who suggested that five specimen replications are required for each test, where the highest and lowest values should be disregarded, and the average of the rest three specimens represents the impact record. However, significant research was conducted during the last few years using the repeated impact test procedure. The ACI 544-2R repeated impact test was used to evaluate the impact performance of several types of cementitious mixtures in the literature, including normal-strength and high-strength concrete [29], steel-fiber-reinforced concrete [30,31,32], synthetic-fiber-reinforced concrete [33,34,35,36,37], hybrid-fiber-reinforced concrete [30,38,39,40], plain and fibrous self-compacting concrete [41,42,43,44,45,46,47], high-performance and ultra-high-performance concrete [48,49,50,51,52], engineered cementitious composites [53,54,55,56], plain and fibrous preplaced aggregate concrete [57,58,59,60,61] and layered concrete [62,63]. On the other hand, other studies tried to investigate the effect of the addition of some materials and additives, such as crumb rubber [41,44,64], recycled materials [42,43,65], natural fibers [6,36], nano material [66,67] and others [68] on the impact performance using this test procedure.

Most of the reviewed experimental works reported very high scattering of results in terms of cracking and failure impact numbers, with noticeably high coefficients of variation (COVs). The high scattering of results urges the use of many more specimens to afford the minimum required degree of confidence [34], which in turn decreases the reliability of the test. However, the repeated impact procedure provides a quick, simplified and low-cost solution for material-scale impact tests. Therefore, it is continued to be used by several researchers for its aforementioned advantages in spite of its high results variation disadvantage. Trials were made by previous and recent research works to modify the test setup, aiming to minimize the effect of the test parameters that lead to the high scattering of results, which would make the ACI 544-2R a more reliable, simple and low-cost alternative.

## 2. Research Significance

According to the best of the authors’ knowledge, there is no previous study that presented a state-of-the art literature review on the repeated impact performance of concrete using the ACI 544-2R procedure. The effect of mixture characteristics on the results variation of the repeated impact test are not yet well explored. The current study provides a brief review about the former and recent studies on the impact tests carried out on different concrete types using the ACI 544-2R procedure. Based on the introduced literature, statistical analyses are introduced to outline the influence of material parameters, including binder, aggregate, water and fiber contents, in addition to others, on the variation of the recorded impact results. Moreover, the literature suggestions to modify the test setup, specimen shape and test procedure to reduce the scattering of the cracking and failure impact results are also reviewed and summarized in this article. The degree of result variation reduction when considering the suggested modifications is also discussed in terms of statistical parameters. By highlighting the high scattering of results and discussing the suggested solutions to overcome this disadvantage, this article is designed to be a useful guideline for future research with the aim of optimizing a reliable, low-cost version of the ACI 544-2R repeated impact test.

## 3. Previous Literature on the Statistical Evaluation of ACI 544-2R Impact Results

The overviewed literature showed that in addition to the standard result dispersion tools such as standard deviation and coefficient of variation, two statistical tests have been used to analyze the repeated impact test results, which are the normal distribution and Weibull distribution tests. Table 1 summarizes most of the previously published works on the repeated drop-weight impact tests where the ACI 544-2R test procedure was adopted. The table focuses on the research works where statistical tools were used to analyze the distribution and variation of the cracking and failure impact numbers. The presented data in Table 1 roughly covers the works conducted during the last 30 years on this topic. To make it easier for readers to follow the presented information in Table 1, a brief literature review is introduced in this section about the listed works.

### 3.1. Normal Distribution

As one of the earlier studies, Soroushian et al. [34] conducted experimental tests to investigate the statistical variation of different mechanical properties of cementitious composites reinforced with carbon fiber (CF). The repeated impact test of ACI 544-2R was one of the investigated tests, where 2% volume fraction of 3 mm long CF was adopted in a mixture that included lightweight aggregate and silica fume in addition to Portland cement. The authors evaluated the variations of the failure numbers both within and between the different patches of specimens, where 30 cylindrical specimens were tested in two patches of 15 specimens each. The results showed that the coefficients of variation (COVs) of the two batches were 36.3 and 39.5%, while the COV between batches (all 30 specimens) was 54.6%. Normal probability distribution was the main statistical evaluation tool, where poor fitness to normal distribution was reported owing to the large variation in test results. Nataraja et al. [32] tested two mixtures of moderately low-strength concrete with and without 0.5% of 27.5 mm length crimped steel fibers (SFs). Thirty-two plain specimens were tested using the ACI 544-2R repeated impact test in two patches of 16 specimens each, while 30 SF-reinforced specimens were tested in two patches of 15 each. The test results revealed poor fitness to normal distribution at a 95% level of confidence for both cracking and failure stages and both mixtures, where the variations within and between patches of both plain and fiber-reinforced specimens were high.

Song et al. [39] conducted ACI 544-2R repeated impact tests on concrete reinforced with SF and concrete reinforced with hybrid fibers of SF and polypropylene fibers (PP). The 28-day compressive strength of both concretes was between 24.2 and 25.6 MPa. The used SF was hooked-end with a length of 40 mm, while 12 mm long PP fibers were used. The adopted volume fractions of SF and PP were 0.5 and 0.1%, respectively. Twenty-four SF-reinforced cylindrical specimens were tested under impact loading, while forty-eight were reinforced with hybrid SF-PP fibers. The test results showed that both concretes exhibited comparable ranges of mean first crack numbers (234 and 247 blows) with a COV range of 54 to 59%. Similarly, the recorded mean failure numbers were very close (330 and 356 blows), with COV ranging from 41 to 52%. Both concretes showed a weak tendency to follow the normal distribution, both at cracking and failure. The reliability of the obtained impact results was also analyzed using the Kaplan–Meier procedure, which indicated a slightly better performance of the hybrid-fibrous mixture at cracking and failure compared to the SF-reinforced mixture. In another study, Song et al. [29] tested the impact performance of plain and SF-reinforced high-strength concrete. Hooked-end SFs were adopted at a volume fraction of 1.0% in the fibrous mixture. The concrete compressive strengths of the plain and fibrous mixtures were 66 and 76 MPa, respectively. For each of the two mixtures, 48 cylindrical specimens were used to evaluate the cracking and failure impact resistances according to the specifications of the ACI 544-2R procedure. The fibrous specimens recorded much higher cracking and failure numbers of blows compared to the plain specimens. For instance, the mean failure numbers of the plain and fibrous mixtures were 454 and 1896 blows, respectively. However, both mixtures exhibited high variations at cracking and failure, with COVs of 42 and 44% at cracking and 41 and 42% at failure. The normal probability test and the Kolmogorov–Smirnov test showed that the plain and fibrous high-strength concrete impact results exhibited poor normal distributions, which revealed the high variation at the cracking and failure stages.

Badr et al. [35] carried out experimental tests to study the statistical variation of impact test results of fiber-reinforced concrete. A normal-strength concrete reinforced with virgin PP fibers (3 kg/m^3^) was tested against impact loading using the ACI 544-2R repeated impact test recommendations. The average compressive strength of the specimens was 41.3 MPa. Two specimen batches with 10 replications each were tested to record the cracking and failure numbers. The means of the cracking and failure numbers of the tested specimens were 57 and 80 blows, respectively. The test results did not fit the normal distribution, where a clear departure from this distribution was observed. The specimens’ records also revealed high result variations, with COV ranges of 56.6 to 61.4% at the cracking stage and 48.7 to 52.1% at the failure stage. Rahmani et al. [36] carried out ACI 544-2R impact tests on concrete reinforced with three different types of fibers. The adopted fibers were 35 mm length hooked-end SF, 12 mm length PP fibers and 2.1 mm length cellulose fibers (CE). The SF was added at a volumetric content of 0.5%, while the volume fraction for PP and CE fibers was 0.15%. To evaluate the cracking and failure impact numbers of the three mixtures, 32 specimens were prepared and tested for each type of fiber. The results were compared with those of a plain mixture that had a similar compressive strength, where the compressive strength of the plain and fibrous mixtures was in the range of 41.8 to 43.2 MPa. The impact test results revealed weak normal distribution of the plain concrete, SF-reinforced concrete and PP-reinforced concrete, while the impact results of the CE-reinforced concrete showed a better tendency to follow the normal distribution compared to the other three mixtures. However, the COV of all mixtures revealed the high scattering of results at cracking and failure, where the COV of the four concretes was in the range of 47 to 65% at the cracking stage, while it was in the range of 39 to 57% at the failure stage.

Mastali et al. [45] conducted repeated impact tests on self-compacting concrete reinforced with recycled-glass-fiber-reinforced polymer (GFRP) at three different volumetric dosages of 0.25, 0.75 and 1.25%, while the cement, silica fume, fine aggregate and water contents were constant for the three fibrous mixtures and a plain reference one. For each of the fibrous mixtures, 40 cylindrical specimens were tested based on the procedure of the ACI 544-2R impact test. The test results revealed approximately similar ranges of result variations for all groups with a COV range of 36.8 to 43.9% for cracking and failure records. As for other types of fibers, the recycled GFRP showed the potential to increase the impact resistance at cracking and failure stages. The test results showed that the cracking test records were better following the normal distribution compared to the ultimate numbers of impact blows. The authors also indicated a high level of scattering of the impact results, where the minimum required number of specimens to keep the error below 10% at a level of confidence of 90% ranged between 23 and 32 replications. Fakharifar et al. [48] adopted the ACI 544-2R impact test recommendations to evaluate the impact resistance of high-performance cementitious composites. The mixtures contained equal high quantities of cement and fine aggregate of 980 kg/m^3^ and were reinforced with 0.5, 0.75 and 1.0% volume contents of 12 mm length PP fibers. For each of the three mixtures, 40 cylindrical specimens were tested until cracking and failure occurred. The test results revealed that both cracking and failure numbers increased with the increase in PP fibers, where the recorded impact numbers at failure increased from approximately 49 to 68 and 81 blows as the PP content increased from 0.5 to 0.75 and 1.0%, respectively. High scattering of results was reported, where the COV of the obtained results ranged from 40 to 47% for cracking and failure records. The statistical analysis of the results showed that compressive and flexural strengths much better fitted the normal distribution compared to the impact records, where the *p*-values of the null hypothesis were much smaller for the impact records. However, they were still higher than the 0.05 limit, which indicates that in spite of the high result dispersion, the impact results hardly followed the normal distribution. Mohammadhosseini et al. [57] investigated the effect of adding waste-metalized plastic fibers (WMPF) on the impact response of concrete, which was investigated in mixtures with or without the partial substitution of 20% of the Portland cement with palm oil fuel ash. The waste plastic fibers were 2 mm wide, 20 mm long and 0.07 mm in thickness. Six mixtures were prepared with fiber contents of 0, 0.25, 0.5, 0.75, 1.0 and 1.25%, while six other, similar mixtures were made with partial substitutions of Portland cement. The results indicated a positive effect of utilizing WMPF in enhancing the impact resistance of concrete. The *p*-values of the obtained impact results (based on the Kolmogorov–Smirnov test) showed that these results hardly followed the normal distribution. Murali et al. [49] conducted repeated impact tests on concrete specimens that included green substitutions to cement and fine aggregate, where two identical mixtures were prepared with or without 0.5% of 50 mm length hooked-end SF. The cement was partially replaced with 2% of nano silica, while 30% of the fine aggregate was replaced with copper slag. Forty cylindrical impact specimens were cast and tested for each mixture following the procedure of the ACI 544-2R repeated impact test. The test results revealed that both cracking and failure records could approximately follow the normal distribution based on the Kolmogorov–Smirnov test. Abid et al. [69] conducted an experimental study using a single high-performance concrete mixture that was adopted with high cement content, silica fume and fly ash composing a total binder quantity of 1160 kg/m^3^, while 960 kg/m^3^ of silica sand was used as the filler of the mixture. The used fiber was 15 mm length straight micro-SF, which was added at a volumetric content of 2.5%. The authors reported that the normal probability distribution was not suitable to describe the distribution of the impact results. Jabir et al. [51] conducted impact tests on six groups of cylindrical specimens made of reactive powder concrete reinforced with SF and SF-PP hybrid fibers. Micro-steel fibers with 6 mm and 15 mm length and PP fibers with 18 mm length were combined at a total volumetric content of 2.5%. The number of test replications in each group was 12 specimens. The results reflected high variations with COV values of 36 to 49%, while all groups failed to follow the normal probability distribution.

### 3.2. Weibull Distribution

Chen et al. [70] tested the impact performance of specimens made of normal concrete and SF-fiber-reinforced concrete with two weight contents of 20 and 30 kg/m^3^. Similar specimens were also made but with steel bar reinforcements, where 12 mm bars were utilized at 50 mm spacing. The cracking and failure impact numbers of the SF-reinforced specimens were several times higher than those of the corresponding plain specimens. On the other hand, the use of steel bars reduced the effect of SF on enhancing the cracking resistance, where minor developments in cracking resistance were recorded compared to the specimens without fibers. However, the failure numbers were extraordinarily raised when steel bars were utilized. The results of the six groups exhibited high variations in the test results, where the COV of the specimens without steel bars ranged from 47 to 75% at the cracking stage and from 33 to 68% at the failure stage. On the other hand, the specimens with steel bars exhibited lower COV values ranging from 23 to 43% in both stages. The authors utilized two-parameter Weibull distribution to statically analyze the obtained impact results, where all impact results exhibited good fits to this distribution with linear regressions, having coefficient of determination (R2) values exceeding 0.94. Ali et al. [54] conducted ACI 544-2R impact tests on plain and fiber-reinforced engineered cementitious composites (ECC). Polyvinyl alcohol (PVA) fibers were used at a constant volume content of 2% with or without shape memory alloy (SMA) fibers at volume fractions of 0.5, 1.0 and 1.5%. The test results revealed the weak impact resistance of the plain ECC, while adding SMA fibers significantly enhanced the impact resistance of the PVA-ECC mixture; the impact numbers increased with the increase in SMA content. It was also reported that Weibull distribution was a good statistical tool to evaluate the impact resistance of the tested specimens.

Abirami et al. [37] utilized the ACI 544-2R impact test to evaluate the cracking and failure impact strengths of four types of concrete mixtures that were reinforced with SF. The four concrete mixtures were fiber-reinforced, two-stage fibrous, layered fibrous and slurry infiltrated fibrous (SIFCON) concretes. In total, nine SF-fibrous mixtures were prepared with different fiber contents ranging from 1.0 to 10.0% and compared with a plain normal concrete mixture. For each of these mixtures, six cylindrical specimens were tested. The test results showed that SIFCON specimens (SF contents of 8 and 10%) exhibited the highest impact records among the nine mixtures, followed by the layered fibrous specimens, while the conventional SF-reinforced mixtures (SF content 1.0 and 1.5%) exhibited the lowest records among the fibrous mixtures, which were several times higher than plain mixture. The authors indicated that the Weibull distribution was a very suitable statistical tool to estimate the impact numbers at cracking and failure stages, where all impact records followed the Weibull distributions, with R2 values of the linear fits exceeding 0.85. Murali et al. [71] tested the impact resistance of two-stage fiber-reinforced concrete (TSFRC), where two types of steel fibers were used, which were short, crimped SF with a 15 mm length and longer, hooked-end SF with a 30 mm length. The fibers were added at three volume dosages of 1.5, 3.0 and 5.0%, composing six fibrous mixtures (three of each type of SF), while a seventh plain reference mixture was prepared for comparison purposes. Fifteen cylindrical test specimens were tested from each mixture following the repeated impact setup of ACI 544-2R. The results showed the use of the two-stage technique enabled a comfortable addition of large quantities of fibers (5%) which increased the impact resistance. The COVs of the test results were recorded to be from 11 to 53% at cracking and from 9 to 30% at failure. The two-parameter Weibull distribution was found to be a good and adequate analysis tool for impact results, where all results followed the Weibull distribution well. Asrani et al. [40] carried out repeated impact tests on geopolymer concrete reinforced with hybrid fibers, where three different types of fibers were utilized. The used fibers were 5D, hooked-end SF with a 50 mm length, 15 mm length glass fiber GF and 13 mm length PP. The fibers were used in mono-fibrous mixtures and hybrid-fibrous mixtures with total fiber volume contents of 0.3, 0.6 and 1.5%, while a plain mixture was used as a reference one. On the other hand, the other ingredients were the same for the plain and fibrous mixtures. Five cylindrical specimens were tested from each mixture using the standard procedure of the ACI 544-2R repeated falling-weight impact test. The test results showed that the inclusion of SF is the main factor that increased the impact resistance, while adding both 0.3% of GF and 0.3% of PP to 1.0% of SF resulted in the highest impact records among all mixtures. The results of the study also revealed that the variations in the test results were higher at the cracking stage than at the failure stage, where the COVs of all mixtures were in the range of 28.6 and 50.6% at cracking and between 14.9 and 45.7% at failure. The results also revealed good agreement with the two-parameter Weibull destruction, where the linear correlations of this distribution for the cracking and failure stages of all mixtures showed high correlations with R2 values ranging from 0.86 to 0.99.

Murali et al. [58] tested the impact performance of three-layered, prepacked aggregate fibrous concrete reinforced with four types of SF. The SF types used were crimped fibers with 14 and 50 mm lengths and hooked-end fibers with lengths of 30 and 60 mm. The fiber content was 3.0% in the top and bottom layers and 1.5% in the middle layer, which means that the average fiber content in the specimens was 2.5%. Glass fiber meshes were used in two layers in between the concrete layers of a group of specimens, while another group was kept without intermediate GFM. Thus, for the four SF fiber types, eight mixtures were prepared with or without intermediate GFM. In addition, two more similar plain mixtures were used as control ones. Six cylindrical specimens were used to evaluate the cracking and failure impact performance of each mixture. The results showed that SF could increase the impact resistance of the mixtures by several times compared to the plain mixture. On the other hand, the use of GFM shared a smaller amount of the total percentage increase in impact resistance compared to SF. The results showed a significant variation at both cracking and failure stages, while the variation was higher at cracking than at failure. The calculated COVs of the 10 mixtures in the cracking stage were 23.6 to 46.2%, while the corresponding COV values at failure were 16.8 to 40.0%. The Weibull distribution was also identified as a good tool to evaluate the statistical distribution of cracking and failure impact results. Haridharan et al. [62] carried out ACI 544-2R impact tests on two-layer grouted-aggregate fibrous concrete (TLGAFC) with intermediate insertions (between the two concrete layers) of glass fiber mesh (GFM) and textile fiber mesh (TFM), which were inserted in different diameters of 0, 50, 75, 100, 125 and 150 mm. A constant content of 5D hooked-end SF of 3% was added for 10 fibrous mixtures, while 10 similar mixtures were plain. The number of mesh insertion layers was also variable with either one or two layers. The results indicated good enhancement in impact resistance, which was attained due to the incorporation of GFM and TFM, which increased with the increase in the mesh diameter and the number of layers. However, the greatest contribution to impact resistance was attributed to steel fibers, which was several times higher than the contribution of intermediate meshes. The calculated COV of the presented data was in the range of 25 to 66% at cracking and 17 to 51% at failure, which also reveals the high variation of impact results. It should be mentioned that six impact cylindrical specimens were tested for each mixture. The results of all mixtures at cracking and failure were found to follow the Weibull distribution well, which was encouraged as a useful analysis tool for this test. The R2 values of the linear fit of all mixtures was approximately in the range of 0.90 to 0.99. Jabir et al. [52] carried out falling-weight repeated impact tests on high-performance fibrous concrete reinforced with SF and PP fibers. Six mixtures were introduced with different hybrid combinations of fibers with a total volumetric content of 2.5%. Twelve identical cylindrical specimens were cast from each mixture to perform the ACI 544-2R impact tests. The tests showed that this type of concrete has high potential to absorb high impact forces before being cracked. The effect of SF was found to be noticeably higher than the effect of PP in increasing the impact resistance, while specimens with longer SF fibers exhibited higher cracking numbers compared to those with shorter SF fibers. The COVs in the cracking stage were in the range of 35.7 to 48.8%, which reflected the high scattering of the impact test results. The test results of all mixtures followed the Weibull distribution, which means that the cracking impact resistance could be evaluated based on the Weibull reliability tables at the required level of confidence. Prasad and Murali [63] tested the repeated impact response of functionally graded preplaced aggregate fibrous concrete (FPAFC), which was placed in molds into single, double or triple layers. Crimped, hooked-end SFs with 50 mm lengths were used in addition to 45 mm length PP fibers. The fibers were used in different combinations to compose 11 different fibrous mixtures in addition to a plain reference mixture. The total (average) content of fibers in the test specimen was 2.4%, which was either constant or distributed differently in each of the two or three preplaced concrete layers. To evaluate the cracking and failure impacts, 15 specimens were tested for each of the 12 mixtures. The results indicated that adding the SF in the top and bottom layers of the three-layer specimens retained higher impact results compared to the single-layer specimens, while the two-layer specimens recorded lower impact strengths compared to the single-layer ones. The results also indicated a significant results variation in the cracking stage, where the COV was calculated to be in the range of 32.4 to 46.2%, while the failure impacts showed lower results variation with a COV less than 43.8. The statistical analysis of the test results also indicated that all cracking and failure impact records followed the Weibull distribution. The goodness of the linear fit revealed high correlations for all cases, where for all mixtures and in both the cracking and failure stages, the R2 of the linear fit of Weibull distribution was in the range of 0.91 to 0.99, except one case with an R2 of 0.88. Other recent researchers also approved the adequacy of Weibull distribution to analyze the repeated impact test results [46,67,72].

**Table 1 materials-15-03948-t001:** Statistical evaluation tools of impact cracking and failure numbers in the literature.

Reference	Concrete Type	Number of Specimens Tested per Mix	Fiber Type	Fiber Content	Compressive Strength (MPa)	COV (%)	Statistical Technique
Soroushian et al. 1992 [34]	Carbon-fiber-reinforced cement composites	30 (two patches of 15)	CF	2%	28.6	36.3–54.6%	Normal probability
Nataraja et a. 1999 [32]	Steel-fiber-reinforced concrete	15	SF	0.5%	29.4–36	46–57.3%	Normal probability
Song et al. 2005 [39]	Hybrid steel-PP-fiber-reinforced concrete	48	SF, PP	0.5% SF 0.1% PP	24.2–25.6	41–59%	Normal probability and Kolmogorov–Smirnov test
Song et al. 2005 [29]	High-strength fiber-reinforced concrete	48	SF	1%	66–76	41–44%	Normal probability and Kolmogorov–Smirnov test
Badr et al. 2006 [35]	PP-fiber-reinforced concrete	20	PP	3 kg/m^3^	41.3	48.7–61.4%	Normal probability
Rahmani et al. 2012 [36]	Plain concrete, fiber-reinforced concrete	32	CE, PP, SF	0.15, 0.15, 0.5%	41.9–43.2	39–65%	Normal probability, Kolmogorov–Smirnov and Kruskal–Wallis test
Mastali et al. 2016 [45]	Glass-fiber-reinforced polymer self-compacting concrete	4 (plain concrete) and 40 for fibrous mixtures	GFRP	0.25, 0.75, 1.25%	50.2–59.2	36.8–43.9%	Normal probability and Kolmogorov–Smirnov test
Fakharifar et al. 2014 [48]	High-performance fiber-reinforced cementitious composites	40	PP	0.5–1.0%	46.1–55.3	40–47%	Normal probability, Kolmogorov–Smirnov. Ryan–Joiner and Anderson–Darling tests.
Murali et al. 2018 [49]	Green, high-performance fiber-reinforced concrete	40	SF	0.5	-	39–48%	Normal probability and Kolmogorov–Smirnov test
Mohammadhosseini et al. 2018 [57]	Fiber-reinforced concrete	3	WMPF	0–1.25%	Different ages	-	Normal probability and Kolmogorov–Smirnov test
Jabir et al. 2020 [51]	Hybrid-fiber-reinforced reactive powder concrete	12	SF, PP	2.5%	75.2–82.8	36–49%	Normal probability
Abid et al. 2020 [69]	Steel-fiber-reinforced high-performance concrete	15	SF	2.5%	81.7	21.2–57.8%	Normal probability
Chen et al. 2011 [70]	Steel-fiber-reinforced concrete	6	SF	0.5%	66.1–67.3	23–75%	Two-parameter Weibull distribution
Ali et al. 2017 [54]	Engineered cementitious composite	3	PVA and SMA	(0, 2.0% PVA) (0.5, 1.0, 1.5% SMA)	-	-	Two-parameter Weibull distribution
Abirami et al. 2019 [37]	Multi-layered grouted fiber-reinforced concrete, Slurry infiltrated fibrous concrete	6	SF	1–10%	34.2–61.8	-	Two-parameter Weibull distribution
Asrani et al., 2019 [40]	Hybrid fibrous geopolymer composites	5	SF, PP, GF	0.3–1.6%	62.4–84.6	14.9–50.8%	Two-parameter Weibull distribution
Murali et al., 2019 [71]	Two-stage fiber-reinforced concrete	15	SF	1.5–5.0%	33.3–51.3	9–53%	Two-parameter Weibull distribution
Jabir et al. 2020 [52]	High-performance fiber-reinforced concrete	12	SF, PP	2.5%	75.2–82.8	35.7–48.8%	Two-parameter Weibull distribution
Murali et al. 2020 [58]	Multi-layered preplaced aggregate fibrous concrete	6	SF	2.5%	33.3–48.5	16.8–46.2%	Two-parameter Weibull distribution
Haridharan et al. 2020 [62]	Multi-layered grouted fiber-reinforced concrete	6	SF	0, 3%	32.4–54.7	17–66%	Two-parameter Weibull distribution
Prasad and Murali 2021 [63]	Preplaced aggregate fibrous concrete	15	SF, PP	2.4%	31.6–50.3	-	Two-parameter Weibull distribution

CE = cellulose fiber, CF = carbon fiber, PP = polypropylene fiber, SF = steel fiber, GF = glass fiber, GFRP = glass-fiber-reinforced polymer, PVA = polyvinyl alcohol, WMPF = waste-metalized plastic fiber, SMA = shape memory alloy fiber.

## 4. Evaluation of Statistical Variation of the ACI 544-2R Repeated Impact Test

The dispersion of the test records of the ACI 544-2R repeated drop-weight impact test results can be attributed to different influential factors. To study the effect of each of these parameters on the dispersion of repeated impact results, sufficient experimental records were collected from the literature. The dispersion analysis was conducted based on the COV values reported by previous works. Therefore, only the research works that reported full details about the obtained data including the COV were adopted from Table 1 for this analysis. The investigated parameters were mostly mix-related variables, which are the compressive strength of concrete, binder content, aggregate–binder ratio, maximum aggregate size, water–binder ratio, fiber volume fraction and fiber length. In addition, the only investigated factor that was related to the test setup was the number of replication specimens. This is because all of the other influential factors of the test setup (such as drop weight, drop height, specimen type and dimensions…etc.) are fixed in the standard ACI 544-2R test, while no standard limitation was proposed by ACI 544-2R for the number of replication specimens. The analysis was conducted for both the cracking and failure numbers, and the degree of effectiveness of each parameter was evaluated based on the correlation degree of the linear fit, as shown in Figure 1, Figure 2, Figure 3, Figure 4, Figure 5, Figure 6, Figure 7 and Figure 8. A strong linear relation between the COV and the investigated parameter with a defined trend of increase or decrease reflects a strong effect of this parameter on the variation of impact results, which is represented by the COV, and vice versa.

Figure 1 explores the influence of the concrete compressive strength on the variability of the ACI 544-2R test results. It is obvious in the figure that the linear relation between the compressive strength and the COV of the cracking impact number is very weak, meaning the coefficient of determination R2 is only 0.019. Similarly, the correlation between the COV of the failure number and the compressive strength is also extremely weak, with an R2 of only 0.004. The weak correlation is also explicitly revealed by the semi-horizontal correlation line, which reflects no specific trend of increase or decrease in the COV with the compressive strength. Thus, it can be concluded that the compressive strength of the mixture has no effect on the variation of cracking and failure impact results.

Figure 2 shows that the content of binder (cement and other cementitious materials) has a minor effect on the dispersion of impact results, which is revealed by the low R2 values of 0.268 and 0.317 in the cracking and failure stages, respectively. However, this effect can be specified as a positive one, where the COV exhibited a slow decrease trend with the increase in the binder content, as shown in the figure. Oppositely, Figure 3 shows that the increase in aggregate content in the mixture compared to the binder content leads to an increase in the variation of the cracking and failure impact results, where the relations between the aggregate/binder ratio and the COV show an increase trend of the COV with the increase in this ratio. However, the low R2 values of these correlations, which are 0.437 at cracking and 0.106 at failure, lowers the dependency on the evaluation of the result variation based on this weak correlation.

Figure 4 shows that the size of aggregate in the mixture has no effect on the variation of impact test results. This is clear from both the weak correlations and the slope of the correlation line, where for both the cracking and failure numbers, the correlation lines are approximately horizontal, which reflects the insignificant role of the maximum size of the aggregate on the COV of the impact results. This insignificant effect is also confirmed by the extremely low coefficients of determination, which are 0.008 at cracking and 0.025 at failure. As for the maximum size of aggregate, Figure 5 explicitly reveals the insignificant role of the water–binder ratio on the COV of the impact results. As shown in the figure, the lines are almost flat, and the R2 values are as low as 0.006 and 0.02 in the cracking and failure stages, respectively.

Figure 6 shows that fiber content is the most influential parameter among the investigated ones. The figure reveals that increasing the fiber content decreases the result variation represented by the COV. The decrease trend of the COV with the increase in fiber content is clear in the cracking and failure stages, while the relatively significant R2 values of 0.47 at cracking and 0.615 at failure confirm the more significant role of the fiber content on the dispersion of the repeated impact results. Oppositely, the semi horizontal lines and low R2 values shown in Figure 7 and Figure 8 reveal the insignificant effect of both fiber length and number of specimens on the variation of impact results, where the R2 values of the fiber length–COV correlations for cracking and failure numbers are 0.009 and 0.019, while those of the number of replication specimens–COV correlations are 0.024 and 0.087, respectively.

## 5. Suggested Modifications to the ACI 544-2R Repeated Impact Test

Aiming to reduce the variability of the ACI 544-2R repeated impact test results, previous researchers introduced some testing modifications. The suggested test modifications were designed to reduce the effect of the parameters that led to the high scattering of the test results.

### 5.1. Previous Literature Works

Badr and Ashour [73] conducted repeated impact tests on concrete patches reinforced with polypropylene fibers. They introduced a list of sources that might lead to the high scattering of the results of the ACI 544-2R repeated impact test. The first distinguished source was the absence of a predefined acceptable cracking criterion, where the test specifications accept any shape of surface visual cracking and along any direction, which might increase the variability of the results owing to the different failure patterns of the test replication specimens. The second source listed by the authors was the adoption of a single central impact point through the central steel ball. As concrete is a heterogeneous material, the central impact point might concentrate the impact load on a stiff gravel particle or a softer cement matrix layer. Thus, the number of impact blows may increase or decrease for each replication specimen based on the type of material localized under the central impact point. The surfacing criterion was the third defined result-scattering source, where the ACI 544-2R allows one to cast the specimens separately or to cut specimens from a standard 150 mm × 300 mm cylinder. The authors also focused on the failure criterion definition of the ACI 544-2R as the fourth possible source of result scattering, where it is required that at least three of the four supporting steel lugs are to be touched by the test specimen fractured parts to consider its failure, which might increase the recorded number of impact blows after the actual failure of the specimen. This might happen if the specimens are heavily reinforced with fibers that prevent the wide crack opening even after the true fracturing. The absence of a clear definition of an acceptable mode of failure was defined as the fifth source of scattering. This is also an important conclusion, where several failure modes were observed by previous researchers. In the literature [47], the specimens reinforced with higher contents of fibers were reported to exhibit a central fracturing zone before continuous surface cracks became visible, where these specimens did not show a complete fracture after hundreds and even thousands of impact blows, which imposes the need of high effort to crack and fail each of the replication specimens [69]. On the other hand, other specimens without or with low fiber content exhibited a simple fracture to two, three or four parts after a few post-cracking impact blows. To reduce the efforts required to crack and fail the test specimens, Zhu et al. [74] suggested the use of U-shape specimens, which were later adopted by Haruna et al. [75] to conduct impact tests for polymerized concrete for runway applications. However, the suggested specimen shape imposed noticeably different load transferring, specimen holding and failure criterion techniques from the standards of ACI 544-2R. On the other hand, an automatic repeated drop-weight impact machine was introduced and facilitated by Al-Ameri et al. [55,56] to conduct the ACI 544-2R impact test, which significantly reduced the effort and time required to conduct the tests.

To overcome the recognized sources of result scattering, Badr and Ashour [73] suggested five modifications. The authors suggested that all replication specimens should either be cast separately or cut from cylinders, which is mostly the same as what is recommended by ACI 544-2R, where it was observed in the literature that researchers depend on one unified method for all specimens. The important modification was to predefine the cracking path and hence control the cracking and failure criterion. To reach this goal, it was suggested to prepare the specimens with two opposite-end, 25 mm triangular notches, as shown in Figure 9a. This modification allowed the authors to suggest an acceptance criterion of cracking and failure of the test specimens. It was determined that the crack should only be accepted if it was aligned through the predefined path along the edges of the opposite triangles, as shown in Figure 9b, while the failure would be defined if the specimen split into two halves along the defined cracking line or this crack was opened widely so that the specimen touched two opposite lugs. On the other hand, the authors suggested using a 50 mm length central line impact load instead of the central concentrated point impact load, which would better control the cracking path and reduce the effect of stress concentration on the scattering of results. Finally, the authors recommended reducing the specimen thickness to 50 mm to accelerate its cracking and failure to reduce the test effort.

Based on the recommendations of Badr and Ashour [73] to modify the current ACI 544-2R testing technique to reduce the scattering of results, Abid et al. [69] conducted experimental research, where testing modifications were suggested for this purpose. As mentioned earlier, among the reasons that lead to the scattering of the results of ACI 544-2R testing technique are that the crack direction is not controlled and neither is the finishing of the top surface. Therefore, the authors suggested using surface-notched specimens to control the crack initiation and propagation direction. They suggested that a 3 mm wide and 5 mm depth surface notch would be suitable enough to determine the path of the specimen surface cracking. As depicted in Figure 10, two surface notch patterns were proposed: line notch and cross notch. Another important reason for the scattering of results was the concentration of the impact load on a single contact point through the steel ball. Therefore, Abid et al. [69] suggested a different loading, surfacing and a load transferring setup to reduce these effects. The suggestion was to use knife-like steel plates that would be placed to receive the impact loads from the falling weight along the notch. Hence, the load would be transferred along a defined line or cross instead of the single central point of the standard steel ball, as shown in Figure 11. Another modification was suggested by the authors to relieve the reaction stresses from the steel base plate by using fine sand bedding beneath the test specimen. Six groups of specimens were tested under six test setup configurations, which included the use of a standard steel ball and the two suggested line and cross load-transferring plates, each with or without the fine sand bedding. Each of the six groups were composed of 15 specimens to evaluate the scattering of the results of the suggested test setups.

Ramakrishnan et al. [76] conducted standard and modified impact tests on five different preplaced aggregate concrete mixtures with different dosages of steel and polypropylene fibers. The authors adopted the test modifications suggested by Abid et al. [69] including the use of line and cross notched specimens with line and cross loading plates. The authors tested the five mixtures using the standard steel ball configuration, the line notch with line loading plate configuration and the cross notch with cross loading plate configuration. Each of the three configurations was considered with three bedding cases. The first was the standard no-bedding case, while in the second and third cases, graded sand and graded gravel were used as bedding materials. Hence, each mixture was tested under nine test setups. It should be noticed here that Abid et al. [69] suggested using a 30 mm thick layer of fine silica sand with a maximum grain size of 0.2 mm, while the maximum sizes of sand and gravel used by Ramakrishnan et al. [76] were 2.36 and 12.5 mm, respectively. Figure 12 summarizes the sources of the scattering of results defined by Badr and Ashour [73] and the suggested solutions by Badr and Ashour and Abid et al. to overcome these causes.

### 5.2. Discussion of Scattering of Results and the Literature Suggestions

The results of the impact tests conducted by Badr and Ashour [73] showed that the COV of the standard ACI 544-2R procedure was at 58.6% in the cracking stage and 50.2% in the failure stage. On the other hand, the COV was noticeably reduced to 39.4% and 35.2% in the cracking and failure stages when the suggested modifications were adopted, which means that a reduction in the scattering of the test results by approximately 30% was verified by using the suggested modified test setup. Myers and Tinsley [77] conducted repeated impact tests on five concrete mixtures with wood and polypropylene fibers using the test setup modifications of Badr and Ashour. The tests were conducted at ages of 7, 14 and 28 days. Considering the 28-day impact results, the first mixture showed significantly lower COV values of 17.9 and 14.9% in the cracking and failure stages, while higher COV values in the range of 21.6 to 41.2% were recorded for the other mixtures. It should also be mentioned that for all mixtures and at all ages, the COV of the impact results in general did not exceed 46%. Most of the obtained impact results reveal that the scattering range agrees with what was reported by Badr and Ashour.

The test results of Abid et al. [69] showed that all suggested configurations reduced the scattering of results compared to the standard setup. However, the configuration of surface line notch with fine sand bedding reduced the COV by more than 60% compared to the standard test setup. Therefore, the required number of specimen replications, at a 90% level of confidence and maximum error of 10%, reduced from 55 specimens for the standard test setup to only 7 specimens for the case of line notch and sand bedding, which is a very encouraging outcome of the suggested modification. Ramakrishnan et al. [76] noticed that the specimens tested under the modified setups recorded noticeably lower COV values compared with the standard test setup, where both notching and bedding decreased the scattering of results at different percentages for the five mixtures. However, comparing the results presented by the authors, it is clear that on average, for all mixtures, the case with line notch and sand bedding resulted in the best scattering reduction results, where the percentage reduction in the COV was in the range of 58 to 66% for the five mixtures, which supports the results of Abid et al. [69]. Murali et al. [49] tested the impact resistance of eleven functionally graded, preplaced concrete mixtures reinforced with hybrid steel–polypropylene fibers using the line notched specimens and line load-transferring plate. Fifteen specimens were tested from each mixture, and seriously low COV values of 5.9 to 16.6% were reported.

## 6. Conclusions

According to the literature works reviewed in this article and the analysis of the scattering of results of previous works, the following summary remarks can be drawn:Although several available test procedures can evaluate the material and structural performance of concrete under impact loads, the ACI 544-2R repeated impact procedure is the simplest to perform and the one that requires the lowest cost and efforts, where no sophisticated data acquisition system and sensors are required to record the force, vibration, deflection, strain or other physical parameters. Instead, only the number of repeated impact blows needed to cause the cracking and failure of the test specimens are required.The main disadvantage of the ACI 544-2R repeated impact test is the high scattering of the test results, where for a batch of specimens, COV values of 30 to 50% are frequent in the cracking and failure stages. The high result scattering is a source of discomfort for engineers who need experimental test results to make their decisions, which reduces the reliability of the current test procedure, where the number of test replications required to afford 90% reliability with 10% acceptable error increases with the increase in COV reaching more than 30 replications.Most of the reviewed literature showed that the repeated impact test results do not agree with the normal distribution or barely follow the normal distribution. On the other hand, the two-parameter Weibull distribution was reported by many previous researchers as a good statistical tool to analyze the high result dispersion of the ACI 544-2R repeated impact test.The statistical analysis of the available literature test results revealed that most of the mixture parameters have no significant effect on the degree of scattering of the results of the ACI 544-2R repeated impact test. The coefficient of determinations of COV with compressive strength, binder content, aggregate content, aggregate maximum size, water–binder ratio, fiber length and number of specimens were in general less than 0.5, which explicitly reveals that there is no correlation between these factors and the scattering of results. However, it was noticed that a weak correlation (R2 = 0.615) showed a tendency of COV to decrease with the increase in fiber content in the mixture.Badr and Ashour [73] identified five sources that cause the high scattering of the results of the ACI 544-2R repeated impact test. These sources can be summarized as: (i) cracks are allowed to occur along any direction, which makes the (ii) definition of accepted failure not suitable, and leads to the (iii) absence of a specific criterion to accept or reject the failed specimens, and (iv) the application of the load on a single central point and (v) the surface treatment of the specimens (cast and toweled or cut) are other result-scattering sources.Badr and Ashour [73] suggested using specimens with triangular edge notches to encourage cracking and hence failure across a specific path along the opposite notches, which could make it easier to define a criterion for accepting the failed specimens. The standard single impact point through the steel ball was replaced by a 50 mm length central line load, which also helped to specify the cracking path. To overcome the defined sources of result scattering, Abid et al. [69] suggested using a 3 mm wide and 5 mm deep surface diagonal notch to specify the cracking and failure path. To ensure that cracking would only occur along the specified path, the surface steel ball was replaced by a knife-like load-transferring plate, while soft sand bedding was suggested to be used beneath the specimens instead of the stiff steel plate to relieve the stress concentration.Badr and Ashour [73] reported that the COV of the impact results was reduced by approximately 30% when the suggested modified procedure was adopted instead of the standard ACI 544-2R procedure, while Abid et al. [69] revealed that the best of the suggested cases was the use of a surface line notch with sand bedding, which led to a significant reduction in the COV by approximately 60% compared to the ACI 544-2R standard procedure. Subsequent researchers confirmed the reduction in the scattering of results when using the suggested test setups of Badr and Ashour and Abid et al.

## Figures and Tables

**Figure 1 materials-15-03948-f001:**
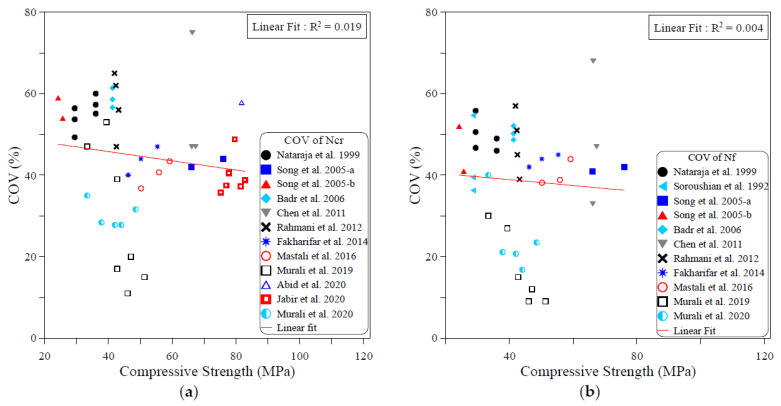
Effect of compressive strength on the scattering of impact results at (**a**) cracking and (**b**) failure.

**Figure 2 materials-15-03948-f002:**
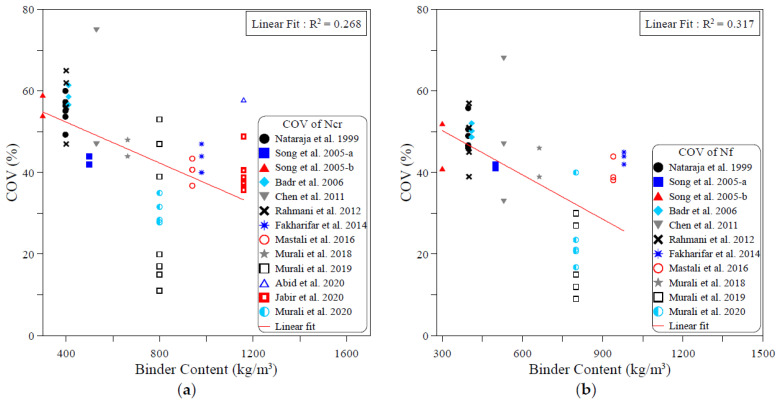
Effect of binder content on the scattering of impact results at (**a**) cracking and (**b**) failure.

**Figure 3 materials-15-03948-f003:**
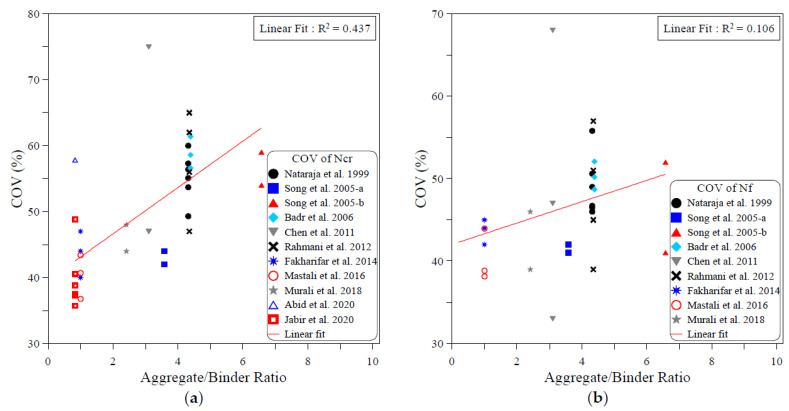
Effect of aggregate content on the scattering of impact results at (**a**) cracking and (**b**) failure.

**Figure 4 materials-15-03948-f004:**
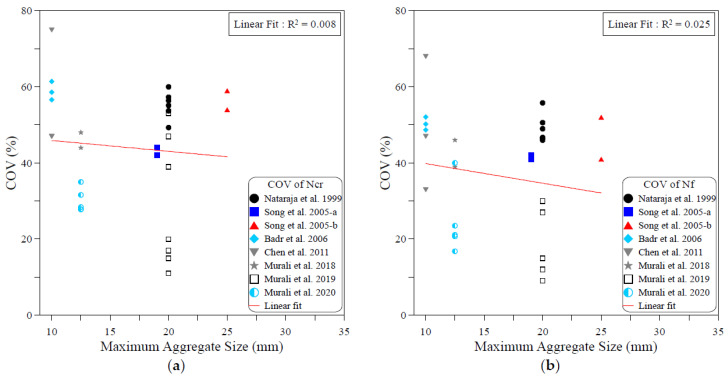
Effect of maximum size of aggregate on the scattering of impact results at (**a**) cracking and (**b**) failure.

**Figure 5 materials-15-03948-f005:**
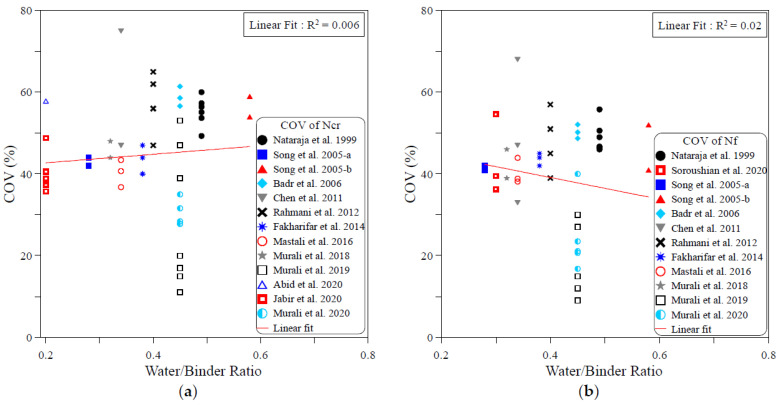
Effect of water–binder ratio on the scattering of impact results at (**a**) cracking and (**b**) failure.

**Figure 6 materials-15-03948-f006:**
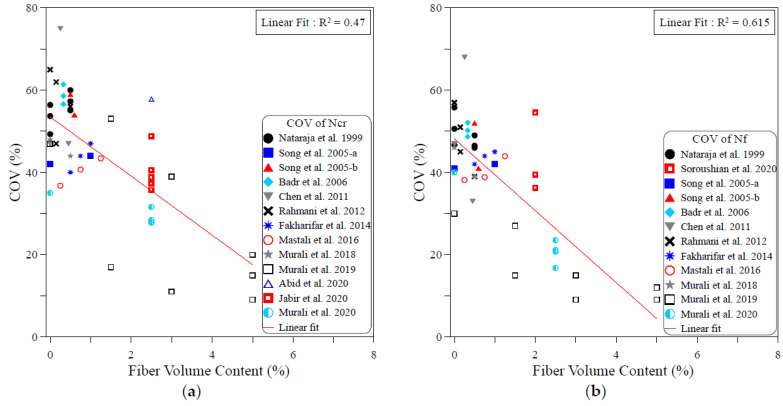
Effect of fiber content on the scattering of impact results at (**a**) cracking and (**b**) failure.

**Figure 7 materials-15-03948-f007:**
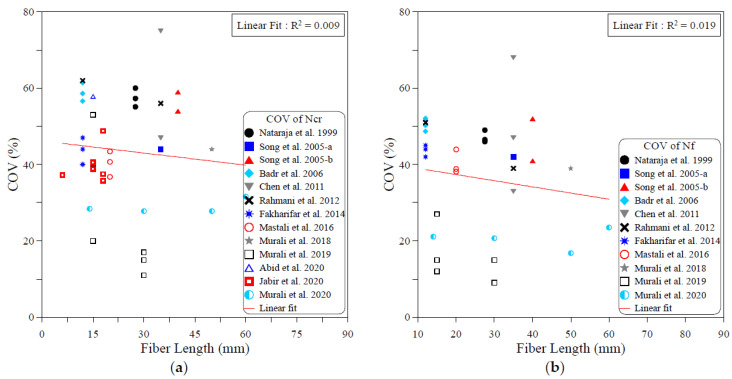
Effect of fiber length on the scattering of impact results at (**a**) cracking and (**b**) failure.

**Figure 8 materials-15-03948-f008:**
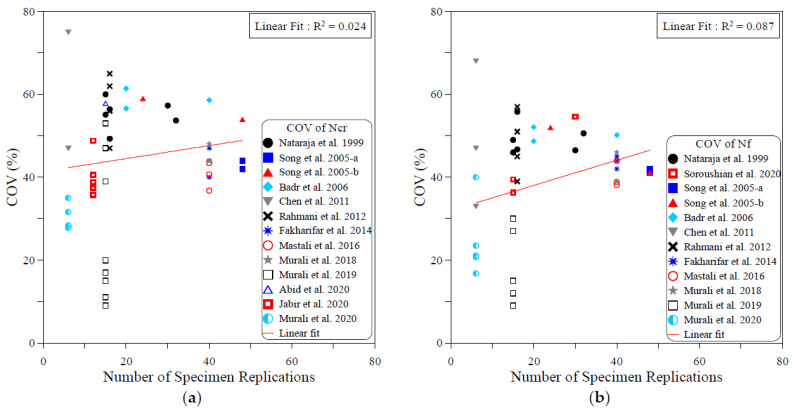
Effect of number of specimen replications on the scattering of impact results at (**a**) cracking and (**b**) failure.

**Figure 9 materials-15-03948-f009:**
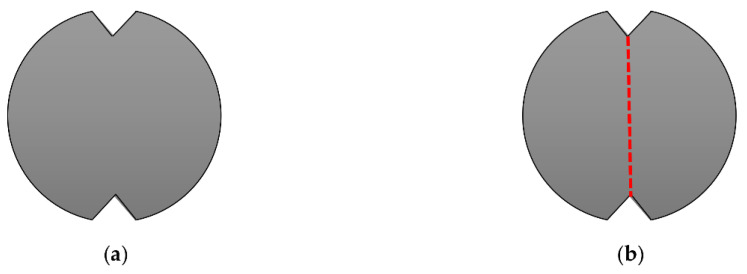
Suggested end-notched specimen of Badr and Ashour [73]: (**a**) end triangular notches and (**b**) accepted cracking path.

**Figure 10 materials-15-03948-f010:**
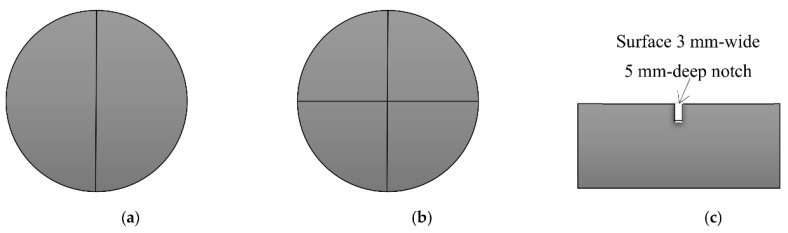
Suggested surface-notched specimen of Abid et al. [69]. (**a**) Surface line notch. (**b**) Surface cross notch. (**c**) Section of the notch.

**Figure 11 materials-15-03948-f011:**
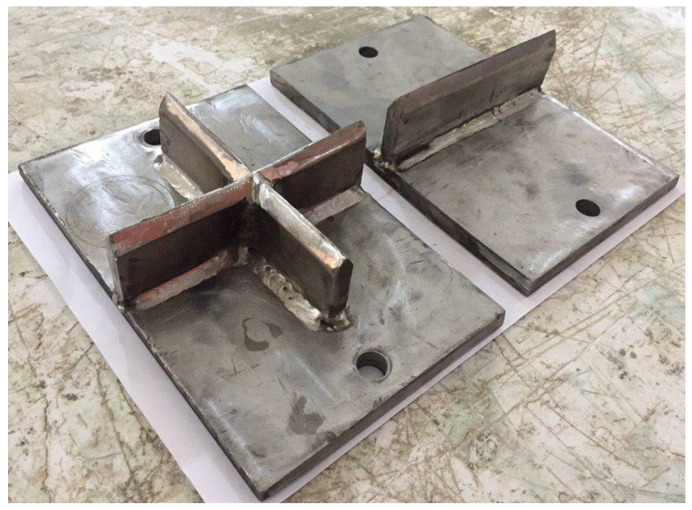
Suggested line and cross load-transferring plates of Abid et al. [69].

**Figure 12 materials-15-03948-f012:**
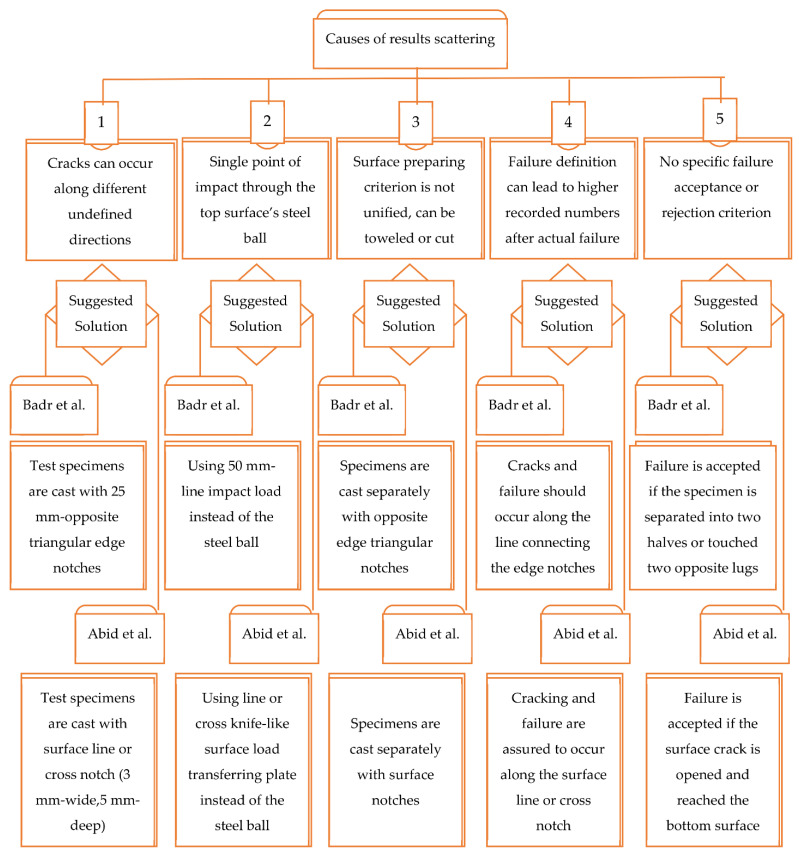
Sources of ACI 544-2R repeated impact scattering of results and suggested modifications by previous studies.

## Data Availability

Not applicable.

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
