# Peer review of "Repeated Drop-Weight Impact Testing of Fibrous Concrete: State-Of-The-Art Literature Review, Analysis of Results Variation and Test Improvement Suggestions"

_materials, 2022, doi:10.3390/ma15113948_

Round 1

Reviewer 1 Report

This paper performed a literature review on repeated drop-weight impact testing of fibrous concrete, and the results variation influenced by different mixture parameters was analyzed. The survey conducted in this study also showed that the test can be modified to lower the unfavorable variations of impact and failure results, and the suggestions were proposed. Generally, this review is comprehensive and the analysis is helpful to improve this testing method. The modification suggestions were listed as below:

1 The discussion should be based on the latest version of ACI 544-2R. The cited standard was established in 1999.

2 What is the criterion of visible surface cracks? This would correlate to the scattered data.

3 The standard ACI 544-2R should be compared with other qualitative tests for evaluating the impact resistance of fibrous concretes under repeated falling-mass impact load.

4 The review on the physical and mechanical mechanism of this test should be added.

Author Response

Reviewer 1:

This paper performed a literature review on repeated drop-weight impact testing of fibrous concrete, and the results variation influenced by different mixture parameters was analyzed. The survey conducted in this study also showed that the test can be modified to lower the unfavorable variations of impact and failure results, and the suggestions were proposed. Generally, this review is comprehensive and the analysis is helpful to improve this testing method. The modification suggestions were listed as below:

  1. The discussion should be based on the latest version of ACI 544-2R. The cited standard was established in 1999.

Response 1: This committee was first introduced in 1989 and was reapproved in 1999; both copies include detailed description about the repeated impact test. The latest copy focuses on new fibrous concretes like fibrous self–compacting concrete and UHPC.

  1. What is the criterion of visible surface cracks? This would correlate to the scattered data.

Response 2: As described in the article, as soon as a surface crack becomes visible (visually checked) on the top surface of the specimen, the specimen is considered cracked and the cracking number is recorded. Actually, as discussed in the article this is one of the reasons of the test’s results high scattering. Therefore, the introduced suggestions focused on the pre-defining of the crack path (Section 5 and Figures 9, 10 and 12).

  1. The standard ACI 544-2R should be compared with other qualitative tests for evaluating the impact resistance of fibrous concretes under repeated falling-mass impact load.

Response 3: The ACI 544-2R repeated impact procedure is the only available standard test recommendation to conduct repeated impact tests on concrete, which was also defined by the committee as a qualitative test as discussed in the article.

  1. The review on the physical and mechanical mechanism of this test should be added.

Response 4: The test procedure is per the recommendations of the ACI 544-2R, which is described in details in the introduction section as highlighted in bright yellow in lines 58-71.

Reviewer 2 Report

The article discuss the important topic of the Repeated drop-weight impact testing of fibrous concrete:  state-of-the-art literature review, analysis of results variation and test improvement suggestions. The article presents very valuable content. In my opinion article should be slightly improved before potential publication. The following modification should be considered:

1. Introduction part is little short and should be rewritten. Consider to contain more literature related to the essence of this studies (different type of concrete, including different type of fibres).
2. I suggest to add separated point - Research significance - Please describe here the main essence of the research. What was the inspiration for such an analysis? Why presented studies are so important?
3. Figure 10 - please improve description.
4. In table 1 please add column with average compressive strength of concrete.
5. It is recommended to clearly indicate potential application of this analysis in civil engineering or another discipline.

Author Response

Reviewer 2:

The article discuss the important topic of the repeated drop-weight impact testing of fibrous concrete:  state-of-the-art literature review, analysis of results variation and test improvement suggestions. The article presents very valuable content. In my opinion article should be slightly improved before potential publication. The following modification should be considered:

  1. Introduction part is little short and should be rewritten. Consider to contain more literature related to the essence of this studies (different type of concrete, including different type of fibres).

Response 1: The introduction is approximately 1000 words long and a separate literature review section of more than 3600 words is already provided in Section 3. However, to satisfy the requirements of this reviewer, a new paragraph was added to describe the most recent literature about this test as highlighted in bright yellow in lines 79 to 90 in the revised manuscript. It should also be noticed that the numbers of most of the cited references were changed accordingly.

  1. I suggest to add separated point - Research significance - Please describe here the main essence of the research. What was the inspiration for such an analysis? Why presented studies are so important?

Response 2: Section 2 was added and included the last paragraph of the introduction that describes the significance of the introduced literature review and scattering analysis. The numbering of the following sections was changed accordingly.

  1. Figure 10 - please improve description.

Response 3: The description was better visualized as required. Please see the highlighted caption on Figure 10(c).

  1. In table 1 please add column with average compressive strength of concrete.

Response 4: A columns was added to include the limits of the compressive strength of each article as highlighted in bright yellow in the revised manuscript.

  1. It is recommended to clearly indicate potential application of this analysis in civil engineering or another discipline.

Response 5: Section 2 (Research significance) well describes why such literature review and statistical analysis is a required study. It shows the significance and the novel points of the introduced study.

Reviewer 3 Report

The state of research on repeated drop-weight impact testing of fibres concrete. Types of testing, specimens, fiber content, fiber length, and specimen size were discussed. Then, analytical, and experimental research developed in the literature to reproduce the response and predict the cracking and ultimate capacity of fiber concrete summarized. However, overall, the manuscript remains rather descriptive and is lacking technicality due to insufficient explanation of the performance of fibers concrete under repeated drop-weight.  According to this Reviewer’s opinion, the paper needs a substantial reconstruction before it could be considered for publication. Please see my comments below:

- The main issue is represented by the fact that the presentation is carried out without distinguishing between the different specimens and testing layouts, which affects the failure mode and performance of the reinforced fiber concrete.

- More analysis and interpretation of the results should be added for a clearer understanding of observed experimental phenomena.

- The authors should provide a discussion on the effect of fiber configuration on the behavior, failure modes, and failure load of concrete specimens.

- Some of the figures provided are unclear and not useful to the reader; therefore, the authors should improve that.

- The English needs to be revised. Typos can be found and some statements are awkwardly constructed and not clear.

-Authors must clarify the necessity of this study. Aims and scope should be presented in the last part of the introduction. (Research significance in presented, but it is not covered the aims and scope of the research).

- Check references and references’ format need to be checked and used the same style throughout the manuscript.

- Add more discussion for figures 1 to 8 and why is it important to report in this study along with other figures?

Author Response

Reviewer 3:

The state of research on repeated drop-weight impact testing of fibres concrete. Types of testing, specimens, fiber content, fiber length, and specimen size were discussed. Then, analytical, and experimental research developed in the literature to reproduce the response and predict the cracking and ultimate capacity of fiber concrete summarized. However, overall, the manuscript remains rather descriptive and is lacking technicality due to insufficient explanation of the performance of fibers concrete under repeated drop-weight.  According to this Reviewer’s opinion, the paper needs a substantial reconstruction before it could be considered for publication. Please see my comments below:

1- The main issue is represented by the fact that the presentation is carried out without distinguishing between the different specimens and testing layouts, which affects the failure mode and performance of the reinforced fiber concrete.

Response 1: For the standard test, the specimens used of all studies the same, which is an approximately 150 mm diameter cylindrical disk with an approximate 64 mm thickness. On the other hand, the suggested configurations by references 69 and 73 to reduce the results scattering were discussed in details, where the configurations were presented (Figures 9 -11) and their effect on the results were discussed (Section 5). As discussed in Section 5, the main aim of the suggested specimen configurations (Figures 9 and 10) is to control the cracking and failure path to reduce the result scattering of the test.  

2- More analysis and interpretation of the results should be added for a clearer understanding of observed experimental phenomena.

Response 2: Results from mostly all available related literature works were collected (Table 1) and detailed statistical analyses on the response of the result scattering, which is the focus of this research, were conducted and discussed using 16 figure (Figures 1 to 8) and approximately 1000 words in Section 4. 

3- The authors should provide a discussion on the effect of fiber configuration on the behavior, failure modes, and failure load of concrete specimens.

Response 3: The authors would like to thank the reviewer for his recommendation. As it is clear in the title and contents of the article, the main focus of this research work is to highlight the deficiency of the high result scattering of the ACI 544-2R test. Therefore, the literature review was focusing on the variation of the results, statistical tools used to analyze this variation and the suggestions to reduce this variation. The analysis presented in Section 4 was also clearly focusing on the effect of mixture and test parameters on this variation. Therefore, discussing other parameters would disturb the presentation of the article and may confuse the readers. It should be mentioned that the authors of this article are now working on writing a literature review that focuses the materials used, cracking and failure patterns and the effect of loading patterns.

4- Some of the figures provided are unclear and not useful to the reader; therefore, the authors should improve that.

Response 4: Thanks for the reviewer for his notice. Figure 10 was improved where a missing caption was better visualized as highlighted in the revised manuscript.

5 - The English needs to be revised. Typos can be found and some statements are awkwardly constructed and not clear.

Response 5: Thanks for the reviewer for his notice. The article was checked for typos and mistakes.

6-Authors must clarify the necessity of this study. Aims and scope should be presented in the last part of the introduction. (Research significance in presented, but it is not covered the aims and scope of the research).

Response 6: Thanks for the reviewer for his notice. A new section (Section 2) was added to highlight the research aims and significance.

7- Check references and references’ format need to be checked and used the same style throughout the manuscript.

Response 7: All references were checked for this issue.

8- Add more discussion for figures 1 to 8 and why is it important to report in this study along with other figures?

Response 8: Figures 1 to 8 discuss the effect of the investigated parameters (material and mixture parameters in addition to test parameters) on the variation of the test results. The discussion was conducted on the basis of the goodness of the linear fit at cracking and failure stages. Therefore, Section 4 provides a detailed discussion on this basis using more than 800 words. 

Reviewer 4 Report

This article aims to review issues related to repeated drop-weight impact testing of fibrous concrete. There are following questions:

  1. Abstract need to be rewritten to report about the main and new findings obtained in this paper briefly.
  2. The conclusion can be more concise.
  3. The authors should emphasize the similarities and differences in the cited articles, and summarize their own views and contributions.

Author Response

Reviewer 4:

This article aims to review issues related to repeated drop-weight impact testing of fibrous concrete. There are following questions:

  1. Abstract need to be rewritten to report about the main and new findings obtained in this paper briefly.

Response 1: Thanks for the reviewer for his notice, the main findings of the presented literature review and analysis are highlighted in bright yellow at the end of the abstract.

  1. The conclusion can be more concise.

Response 2: Thanks for the reviewer for his notice, the main focus of this research work is to highlight the deficiency of the high result scattering of the ACI 544-2R test, and to analyse the effect of the mixture and test parameters on this scattering. Finally, the article presents and discusses the suggestions provided by literature works to reduce this scattering. Based on the conducted analyses, the most important conclusions were summarized in the presented seven conclusion remarks.

  1. The authors should emphasize the similarities and differences in the cited articles, and summarize their own views and contributions.

Response 3: All cited references are of direct relation with the presented literature review and analysis and were properly cited. The cited references almost represent of all the available trusted international research work on the ACI 544-2R test. Most of the up-to-date literature works were cited and analyzed, and their presentation and collection is of great value for interested readers as it highlights the developments in this field of study.

Round 2

Reviewer 3 Report

 The author's addressed reviewer’s comments and as a result the reviewer suggests the paper to be considered for publication